# Useful High-Entropy Source on Spinel Oxides for Gas Detection

**DOI:** 10.3390/s22114233

**Published:** 2022-06-01

**Authors:** Takeshi Hashishin, Haruka Taniguchi, Fei Li, Hiroya Abe

**Affiliations:** 1Faculty of Advanced Science and Technology, Kumamoto University, Kumamoto 860-8555, Japan; 2Faculty of Engineering, Kumamoto University, Kumamoto 860-8555, Japan; h.taniguchi@takada.co.jp; 3Joining and Welding Research Institute, Osaka University, Osaka 567-0047, Japan; h-abe@jwri.osaka-u.ac.jp

**Keywords:** cation, spinel, high-entropy oxides, gas detection

## Abstract

This study aimed to identify a useful high-entropy source for gas detection by spinel oxides that are composed of five cations in nearly equal molar amounts and free of impurities. The sensor responses of the spinel oxides [1# (CoCrFeMnNi)_3_O_4_, 2# (CoCrFeMnZn)_3_O_4_, 3# (CoCrFeNiZn)_3_O_4_, 4# (CoCrMnNiZn)_3_O_4_, 5# (CoFeMnNiZn)_3_O_4_, and 6# (CrFeMnNiZn)_3_O_4_] were evaluated for the test gases (7 ppm NO_2_, 5000 ppm H_2_, 3 ppm NH_3_, and 3 ppm H_2_S). In response to NO_2_, 1# and 2# showed p-type behavior while 3–6# showed n-type semiconductor behavior. There are three p-type and one n-type AO structural compositions in AB_2_O_4_[AO·B_2_O_3_] type spinel, and 1# showed a stable AO composition because cation migration from site B to site A is unlikely. Therefore, it was assumed that 1# exhibited p-type behavior. The p-type behavior of 2# was influenced by Cr oxide ions that were present at the B site and the stable p-type behavior of zinc oxide at the A site. The spinel oxides 3# to 6# exhibited n-type behavior with the other cationic oxides rather than the dominant p-type behavior exhibited by the Zn oxide ions that are stable at the A site. In contrast, the sensor response to the reducing gases H_2_, NH_3_, and H_2_S showed p-type semiconductor behavior, with a particularly selective response to H_2_S. The sensor responses of the five-element spinel oxides in this study tended to be higher than that of the two-element Ni ferrites and three-element Ni-Zn ferrites reported previously. Additionally, the susceptibility to sulfurization was evaluated using the thermodynamic equilibrium theory for the AO and B_2_O_3_ compositions. The oxides of Cr, Fe, and Mn ions in the B_2_O_3_ composition did not respond to H_2_S because they were not sulfurized. The increase in the sensor response due to sulfurization was attributed to the decrease in the depletion layer owing to electron sensitization, as the top surface of the p-type semiconductors, ZnO and NiO, transformed to n-type semiconductors, ZnS and NiS, respectively. High-entropy oxides prepared using the hydrothermal method with an equimolar combination of five cations from six elements (Cr, Mn, Fe, Co, Ni, and Zn) can be used as a guideline for the design of high-sensitivity spinel-type composite oxide gas sensors.

## 1. Introduction

Setting a threshold between the stoichiometric and nonstoichiometric compositions of materials, crystalline and amorphous structures, stable and unstable energies, etc. is challenging. The composition of matter can classify all substances that deviate from stoichiometric composition as non-stoichiometric, crystal structure, crystalline phases, and energy introduce the separate concepts of quasicrystal, intermediate phase, and metastable, which play intermediary roles, respectively. The structures of the high-entropy oxides (HEOs) considered in this study are between crystalline (ordered) and amorphous (disordered); therefore, energy stabilization can be ensured by increasing the entropy. In 2004, high-entropy materials were developed by stabilizing nearly equimolar mixtures and maximizing the coordination entropy of ionic crystals composed of cations and anions in the crystal structure [1,2]. In early studies, metal alloys and nitride thin films were fabricated, and in 2015, entropy stabilization was demonstrated in oxide mixtures [3]. Subsequently, other high-entropy ceramics have been studied with more components added, resulting in materials that represent mixtures with significantly enhanced properties [4].

A previous study investigated the reversible lithium storage properties of HEOs, the underlying mechanisms governing these properties, and the effect of entropy stabilization on their electrochemical behavior [5]. It was discovered that entropy stabilization provided significant advantages in storage capacity retention for HEOs and considerably improved cycling stability. Furthermore, it was confirmed that the electrochemical behavior of the HEOs depended on each of the metal cations present, facilitating the tuning of electrochemical properties by simply changing the elemental composition. Subsequently, a high-entropy source, (Co, Cr, Fe, Mn, Ni)_3_O_4_ oxides, characterized by an Fd-3m single-phase spinel structure was reported for the first time [6]. Thereafter, a new class of high-entropy spinel oxides, (Cr_0.2_Fe_0.2_Mn_0.2_Ni_0.2_Zn_0.2_)_3_O_4_ nanocrystalline powder, was prepared via solution combustion synthesis [7]. These HEOs exhibited ferromagnetic properties when either magnetic Co^2+^ or Ni^2+^ was replaced with nonmagnetic Zn^2+^, drawing attention to the functional properties of HEOs.

The application of spinel oxides in gas sensors as a function of HEOs was reported as early as 1999 when a 1 wt% Pd-loaded p-type semiconductor oxide (NiFe_2_O_4_) was used to selectively detect 1000 ppm of Cl_2_, three orders of magnitude lower than 100% O_2_, liquid petroleum gas (LPG), or CH_4_ [8]. The reason for this outcome is not clear but is suggested to be related to the Fe state. In the inverse spinel Fe[NiFe]O_4_ structure, Fe occupies half of the A (tetrahedral) and B (octahedral) sites, while Ni is mainly stable at the B site, suggesting that the electronic interaction between Cl_2_ and Fe is sensitized by Pd and affects the electrical conduction of the base material, NiFe_2_O_4_. In a study on the response properties of four ferrite MFe_2_O_4_ (M = Cu, Zn, Cd, Mg) samples prepared via coprecipitation with CO, H_2_, LPG, EtOH, and C_2_H_2_, MgFe_2_O_4_ and CdFe_2_O_4_ showed selective responses to LPG and C_2_H_2_, with MgFe_2_O_4_ showing the highest response to LPG at 250 °C [9]. Thick films composed of nanocrystalline MgFe_2_O_4_ were exposed to reducing gases, such as CH_4_, H_2_S, LPG, and EtOH, and the conductance responses were measured, showing selective responses to H_2_S at 200 °C and ethanol at 350 °C [10]. Additionally, polycrystalline MgFe_2_O_4_ prepared through the coprecipitation method showed a selective response to gasoline and LPG at 250 °C [11]. Recently, the magnetic properties and ozone gas-sensing activity of ZnFe_2_O_4_ nanoparticles (NPs) were demonstrated using experimental procedures and density functional theory (DFT) calculations [12]. O_3_ gas sensing has been explained based on conduction changes on the ZnFe_2_O_4_ surface and an increase in the electron depletion layer height due to exposure to the target gas. Based on the experimental results from our previous study, we proposed that the depletion layer may be responsible for the high gas sensing sensitivity, suggesting that the p–n junction effect between p-type MgO and n-type MgFe_2_O_4_ appears in response to 3 ppm H_2_S at 200 °C [13]. To provide evidence for this, the potential barrier at the interface of core-shell microspheres composed of p-MgO/n-MgFe_2_O_4_/Fe_2_O_3_ was visualized using Kelvin probe force microscopy (KPFM) to determine the enlargement of depletion layer due to the p–n junction effect [14]. As described above, spinel oxides are promising sensitizers for gas detection.

In this study, we focused on high-entropy spinel oxides as sensitive materials for gas detection. The objective of this study was to identify a useful high-entropy source for constructing spinel structures for gas detection. Six different HEOs were prepared from six different cations (Co, Cr, Fe, Mn, Ni, and Zn) in combination with five different cations using the hydrothermal method, and the response properties of the HEOs to the test gases (7 ppm NO_2_, 5000 ppm H_2_, 3 ppm NH_3_, and 3 ppm H_2_S) were investigated.

## 2. Materials and Methods

### 2.1. Preparation of HEOs

The starting materials for the preparation of transition metal cations in HEOs, that is, Co(NO_3_)_2_·6H_2_O, CrCl_3_·6H_2_O, Mn(NO_3_)_2_·6H_2_O, Ni(NO_3_)_2_·6H_2_O, and ZnCl_2_ were purchased from Fujifilm Wako Pure Chemical Co. (Osaka, Japan); potassium acetate (KOAC, 98.5% purity), FeCl_3_·6H_2_O, and ethylene glycol (EG, 99.5% purity) were purchased from Kishida Chemical Co. (Osaka, Japan). All chemicals were of analytical grade and used as received. All six metal salts were dissolved in EG to form a 0.2 M solution (abbreviated as M/EG solution). KOAC was dissolved in EG (1.2 M; denoted as the KOAC/EG solution). Five out of the six metal elements were selected to have six equimolar five-element high-entropy compositions, as follows: 1# (CoCrFeMnNi)_3_O_4_, 2# (CoCrFeMnZn)_3_O_4_, 3# (CoCrFeNiZn)_3_O_4_, 4# (CoCrMnNiZn)_3_O_4_, 5# (CoFeMnNiZn)_3_O_4_, and 6# (CrFeMnNiZn)_3_O_4_.

As a typical representative procedure for the synthesis of the aforementioned HEOs, five equimolar metal solutions (2 mL each, 10 mL in total) were mixed with 10 mL of KOAC/EG solution and sealed in an autoclave. The autoclave was heated at 473 K for 7.2 ks and allowed to cool naturally to ca. 298 K. The product was collected, washed thrice with deionized water and ethanol via centrifugation, and dried overnight at 323 K in a vacuum oven. Finally, the products were sintered at 873 K for 7.2 ks for the crystallization of HEOs. Inductively coupled spectroscopy (ICP) analysis confirmed that the five constituent elements of these six HEOs are present in near-equimolar amounts [15].

### 2.2. Identification of the Product Phase of HEOs

X-ray diffraction (XRD; Cu-Kα, 30 kV and 10 mA; Bruker D2, Germany) was used to identify the product phases of the HEOs. From the peaks in the XRD patterns, the crystallite size was calculated using Scherrer’s formula as shown in Equation (1), and the average value was taken for each sample from 1# to 6#.
D = Kλ/βcosθ(1)

D: crystallite size (nm), K: Scherrer constant (0.94) [16], λ = Cu Kα (0.1542 nm), β = full width at half maximum (rad), θ = Bragg’s angle (rad)

### 2.3. Fabrication of Sensing Elements

Each of the six samples was milled for 300 s using a mortar and pestle to prepare a 5 mass% suspension with pure water as the solvent. The samples were then dispersed in the solvent using an ultrasonic cleaner (VS-100III, AS ONE Co., Osaka-shi, Japan; 28 kHz, 60 s; 45 kHz, 60 s; 100 kHz, 60 s).

Screen printing with a gold paste was used to fabricate comb-shaped gold electrodes on alumina substrates that were then heat-treated at 323 K for 600 s (temperature increase rate: 10 K/60 s) and 1073 K for 600 s (temperature increase rate: 10 K/60 s) using an electric furnace. A gold wire was wrapped around the holes at both ends of the finished substrate and the gold paste was applied. The temperature was raised to 923 K (3 K/60 s) using a muffle-type electric furnace (KDF-S70, Denken Co., Ltd., Japan) and maintained for 1.8 ks to remove the solvent component of the gold paste to fabricate the sensor substrate.

A micropipette was used to drop 9 μL of the suspension onto the gold electrode of the sensor substrate. Subsequently, the sensor element was sintered (773 K, 10.8 ks) in a muffle-type electric furnace (KDF-S70, Denken Co., Ltd., Japan).

### 2.4. Gas Detection System

The direct current (DC) circuit shown in Figure 1 was used to measure the resistance and response of the sensor element. A DC-stabilized power supply (PA18-3A, Kenwood Co., Japan) and a digital multimeter were used in a series circuit to measure the voltage change applied to both ends of a reference resistor connected in series with the sensor element. A mass flow controller (MFC; SEC-400, HORIBA, Ltd., Japan) was used to adjust the flow rate of air for resistance measurements as well as the flow rate of air, test gases (NO_2_, H_2_S, NH_3_, and H_2_), and O_2_ gas for sensor measurements. The operating temperature was regulated using a digital program controller/setter (KP1000, CHINO Co., Japan) in a small electric furnace (ARF-30K, Asahi-rika Co., Ltd., Japan). The output voltage was measured and recorded using a digital electrometer (R8240, ADVANTEST Co., Japan) and an Excel (Microsoft) add-in software (W32-R8240, Systemhouse sunrise Co., Japan) via a general-purpose interface bus (GPIB) controller (GPIB-USB-HS, National Instruments Co., USA), respectively, and the data were saved on a personal computer.

The output was measured using the formula shown in Equation (2). The sensor element was attached to a sensor holder; the applied voltage, E, was 1 V, and the resistance, R, of the sensor element was calculated from the voltage, V, applied to the reference resistance using Equation (2). The output voltage, V, was measured by allowing air to flow through the measurement holder (300 s). The average resistance, Ra, is the average resistance estimated in air from 0 to 300 s using Equation (2), and it was calculated at each operating temperature.
R = ((E/V) − 1)r(2)

R: Resistance of the sensor element, E: Voltage of the power supply, V: Voltage at both sides of the standard resistance, r: Value of the standard resistance.

The V of the test gas was measured by allowing the gas to flow through the sensor holder in the following order: air (300 s), test gas (300 s), and air again (300 s). The operating temperatures were 573, 523, 473, 423, 373, and 323 K.

In the case of a p-type semiconductor, the maximum resistance after the flow of the gas to be tested is defined as Rg, and in the case of an n-type semiconductor, the minimum resistance after the flow of the gas to be tested is defined as Rg. To ensure that the sensor response, S, is higher than or equal to 1, S is calculated using Equations (3) and (4) when the test gas is a reducing gas and an oxidizing gas, respectively.
S = (Rg − Ra)/Ra(3)
S = (Ra − Rg)/Rg(4)

## 3. Results and Discussion

### 3.1. Identification of the Product Phase of HEOs

Figure 2 shows the XRD patterns of the six HEOs. All the patterns were attributed to the spinel structure with Fd-3m space groups, and no other crystalline phases were present. The crystallite sizes of 1–6# calculated using Scherrer’s formula, are listed in Table 1. The crystallite sizes of 6#, 3#, 2#, 5#, 4#, and 1# were 28.2, 35.4, 77.0, 79.2, 84.8, and 98.2 nm, respectively, in the decreasing order. All the nanoparticles showed isotropic diffraction patterns.

### 3.2. Electrical Properties of HEOs and Sensing Performance in Response to NO_2_, H_2_, NH_3_, and H_2_S

#### 3.2.1. Electrical Properties of HEOs

The resistance–temperature properties of the six HEOs in air are shown in Figure 3a. The slope of the approximate straight line in Figure 3a is shown in Figure 3b, and the Ra at 523 K is shown in Figure 3c; the larger the slope of the linear approximation, the higher the Ra. The HEOs with high resistivity were 2#, 5#, and 6# while those with low resistivity were 1#, 3#, and 4#. Although the XRD patterns in Figure 2 are similar, it cannot be determined whether the structure is a normal or reverse spinel. In normal (AB_2_O_4_) and inverse (B[AB]O_4_) spinels, the valence of the A and B cations are 2+ and 3+, respectively. To determine whether the five cations in a single HEO are in the normal or inverse spinel structure, the crystal field stabilization energies (CFSE) of the ions, mainly for the tetrahedral and octahedral sites, were used [17]. The relationship between the cationic species and occupancy factor, λ, for spinel A and B sites extracted from [17] is summarized in Table 2. λ = 0 represents normal spinel; λ = 0.5 represents inverse spinel, and the others represent a mixture of normal and inverse spinels.

For example, if the A and B sites are Mn^2+^ and Fe^3+^, respectively, a more normal spinel structure tends to exist. However, the value of λ varies with temperature, and this variation should be considered a guideline. Among the six types of HEOs considered in this study, only 4# without Fe had normal spinels because they were determined to have Cr^3+^ at the B site. The other five types are considered to have a mixture of normal and inverse spinels because it is unclear what proportion of the 0.2 mol% Fe is distributed between the A and B sites. The octahedral site stabilization energy (OSSE) was calculated to confirm the trend of charge migration from the B (octahedral) site to the A (tetrahedral) site [18,19]. The trend of cation migration is indicated by its relationship with the number of 3d electrons. For the six cations in this study, the 3d electron numbers of d3 to d8 were considered as migration levels between high and low spins, and the migration levels were classified as high, medium, and low, and are listed in Table 3a. In addition, the migration levels of the six HEOs were classified as high, medium, and low, as shown in Table 3b, and the percentages of high and medium states were calculated to be 0% for 1#, 20% for 5#, and 25% for others. Therefore, cation migration did not affect the electrical resistance of the six HEOs in this study.

#### 3.2.2. Sensing Performance in Response to NO_2_, H_2_, NH_3_, and H_2_S

Figure 4a–d show the temperature dependence of the sensor response for samples 1–6# to the test gases (7 ppm NO_2_, 5000 ppm H_2_, 3 ppm NH_3_, and 3 ppm H_2_S). For the oxidizing gas, 7 ppm NO_2_ (Figure 4a), 1# and 2# showed a decreased resistance, corresponding to the behavior of a p-type semiconductor, while 3#, 4#, 5#, and 6# showed an increased resistance, which is the behavior of an n-type semiconductor. Among the oxides with AO structural compositions, the p-type semiconductors are FeO [20], CoO [21], NiO [21], and ZnO [22], and the n-type semiconductor is MnO [23]. In oxides with a B_2_O_3_ composition, Co_2_O_3_ [24] and Ni_2_O_3_ [25] as p-type semiconductors and Cr_2_O_3_ [26], Fe_2_O_3_ [21], and Mn_2_O_3_ [27] as n-type semiconductors have been reported. As presented in Table 3b, in 1#, the AO composition is stable because cation migration from the B site to the A site is difficult, and the three p-type AO compositions and one n-type AO composition exhibit p-type behavior. In 2#, except for Ni ions, the AO compositions of FeO, CoO, and ZnO behave as p-type and MnO as n-type oxides, while the B_2_O_3_ compositions of Co_2_O_3_ are p-type, while Cr_2_O_3_, Fe_2_O_3_, and Mn_2_O_3_ are n-type. The Fe ions in Ni–Zn ferrite, a mixture of normal and inverse spinels are present at the A and B sites [28]. As presented in Table 3b, cation migration from the B to A sites in 2# is unstable for the four cations (Cr, Mn, Fe, and Co), whereas these cations are stable at site B. The p-type behavior of 2# can be attributed to the p-type behavior of Cr ions that can exist at site B due to oxides of Cr ions and the stable p-type behavior of ZnO at site A. In 3–6#, the oxides that behaved as n-type in the other cationic oxides were more dominant than the p-type behavior of the oxide of Zn ions (ZnO), which was stable at site A.

Figure 4b–d show the sensor responses to reducing gases (5000 ppm H_2_, 3 ppm NH_3_, and 3 ppm H_2_S). All samples from 1# to 6# showed an increase in resistance and p-type semiconductor behavior at operating temperatures of 323 to 573 K. The sensor responses to 5000 ppm H_2_ and 3 ppm NH_3_ tended to be higher at higher temperatures except for 5#. This may correspond to the fact that carrier migration is enhanced at higher temperatures, and Ra is decreased, as shown in Figure 3. Data in Table 3b indicate that the stability of Zn ions at the A site may have contributed to the p-type behavior of 5# at 523 K because of the absence of Cr ions. The sensor responses to 3 ppm H_2_S were higher at lower temperatures for 2#, while 1#, 3#, 4#, and 6# showed high sensor responses at 423 K, and 5# showed the highest sensor response at 523 K.

The sensor responses for HEOs 1–6# per crystallite size are shown in Figure 5 as a bar graph for each operating temperature. The sensor responses were particularly high for 3# and 6#, and the responses were similar for other test gases (NO_2_, H_2_, and NH_3_). To investigate whether HEOs 1–6# exhibit gas selectivity, the sensor responses (|Rg − Ra/Ra × 100|) at 373 K are shown in Figure 6, and all HEOs showed selectivity for H_2_S. Ni–Zn ferrite (Ni_0.6_Zn_0.4_Fe_2_O_4_) was reported to exhibit the largest sensor response to 50 ppm H_2_S at 498 K [28]. The Ni–Zn ferrite has Fe and Zn ions at the A site and Ni and Fe ions at the B site, and the chemical formula can be expressed as Fe_0.6_Zn_0.4_[Ni_0.6_Fe_1.4_]O_4_.

Figure 7 shows the relationship between the sensor response (%) and H_2_S concentration for spinel oxides at 423 K. The pink, bright blue, and black arrows indicate the sensor response for 6# HEO, Fe_1−x_Zn_x_[Ni_1−x_Fe_1+x_]O_4_ [28], and Fe[NiFe]O_4_ [28] to 50 ppm H_2_S, respectively. The sensor response tends to be higher when the number of cations in sites A and B is higher because high sensor responses can be achieved in spinel oxide-based gas sensors by increasing the number of cation species used in the HEO. Based on the linear fitting of Figure 7, limit of detection (LOD) of 6# HEO was 0.34 ppm.

#### 3.2.3. Dominance of HEO Cations in H_2_S Detection

To identify the cation species that contribute to the high sensor responses to H_2_S, the reactivity of H_2_S with cations was examined separately for AO and B_2_O_3_ compositions using the temperature dependence of the Gibbs free energy (Ellingham diagram) based on the thermochemical equilibrium theory.

The Ellingham diagram for the AO composition is shown in Figure 8a, assuming that the following reactions occur:H_2_S[g] + MnO[sl] = H_2_O[g] + MnS[sl]
H_2_S[g] + FeO[sl] = H_2_O[g] + FeS[sl]
H_2_S[g] + CoO[sl] = H_2_O[g] + CoS[s]
H_2_S[g] + NiO[sl] = H_2_O[g] + NiS[sl]
H_2_S[g] + ZnO[s] = H_2_O[g] + ZnS[s]

A prepared Ellingham diagram assuming the following reactions for the B_2_O_3_ composition is shown in Figure 8b.
1.2H_2_S[g] + 0.50Cr_2_O_3_[s] = 1.2H_2_O[g] + 0.20SO_2_[g] + CrS[s]
1.2H_2_S[g] + 0.50Mn_2_O_3_[s] = 1.2H_2_O[g] + 0.20SO_2_[g] + MnS[sl]
1.2H_2_S[g] + 0.50Fe_2_O_3_[s] = 1.2H_2_O[g] + 0.20SO_2_[g] + FeS[sl]

The AO compositions exhibited spontaneous sulfurization in the calculated temperature range (300–700 K), whereas among the B_2_O_3_ compositions, Cr_2_O_3_ was not sulfurized. Furthermore, the sulfurization degrees of MnO and Mn_2_O_3_ were approximately the same in Figure 8a,b. Among the AO compositions, ZnO and NiO with negative and large Δ*G* dominated the sulfurization. Similar to the high sensitivity due to the p–n junction described in the introduction [13,14], i.e., the decrease in the height of the depletion layer owing to the sensitization of the top surface to electrons, due to sulfurization, the p-type semiconductors, ZnO and NiO, became n-type semiconductors, ZnS and NiS, which may contribute to their increased sensor responses.

These results suggest that HEOs, prepared using the hydrothermal method, which involves combining five cations from six elements (Cr, Mn, Fe, Co, Ni, and Zn) in equimolar amounts, can be used as a guideline for designing highly sensitive spinel-type composite oxide gas sensors.

## 4. Conclusions

HEOs composed of five cations in near-equimolar amounts and free of impurities [1# (CoCrFeMnNi)_3_O_4_, 2# (CoCrFeMnZn)_3_O_4_, 3# (CoCrFeNiZn)_3_O_4_, 4# (CoCrMnNiZn)_3_O_4_, 5# (CoFeMnNiZn)_3_O_4_, and 6# (CrFeMnNiZn)_3_O_4_] were prepared hydrothermally. The sensor responses for these six HEOs to an oxidizing gas of 7 ppm NO_2_ were investigated. HEOs 1 # and 2 # showed p-type behavior, while HEOs 3–6# showed n-type semiconducting behavior. HEO #1 was less likely to transfer cations from the B site to the A site, and the stable AO composition was considered to dominate the conductivity. The oxides with AO compositions of 1# HEOs include three p-type and one n-type composition; therefore, #1 HEOs are considered to exhibit p-type semiconductor behavior. In #2 HEOs, the stable p-type behavior of Cr oxide ions that could be present at the B site and ZnO at the A site affected the conductivity. In HEOs 3–6#, the n-type behavior of the oxides of other cations was dominant over the p-type behavior of the stable Zn oxide ions (ZnO) at the A site, which was considered dominant. In contrast, the sensor responses to the reducing gases 5000 ppm H_2_, 3 ppm NH_3_, and 3 ppm H_2_S indicated p-type semiconductor behavior for all six HEOs; in particular, the response to H_2_S was selective. The sensor responses for the five-element spinel oxides in this study tended to be higher than those of previously reported two-element Ni ferrites and three-element Ni–Zn ferrites. The thermodynamic equilibrium theory of AO and B_2_O_3_ compositions was used to examine the susceptibility to sulfurization, and the results showed that oxides of Cr and Fe ions with a B_2_O_3_ composition were less susceptible to sulfurization. The improvement in the sensor responses via sulfurization is considered to be due to the decrease in the depletion layer height owing to electron sensitization because the top surfaces of ZnO and NiO, which are p-type semiconductors, transformed to ZnS and NiS, which are n-type semiconductors. HEOs prepared via thermal methods may serve as design guidelines for fabricating high-sensitivity spinel-type composite oxide gas sensors.

## Figures and Tables

**Figure 1 sensors-22-04233-f001:**
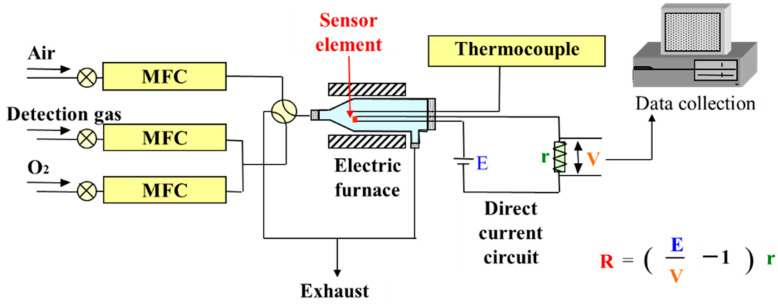
Schematic of the gas detection system.

**Figure 2 sensors-22-04233-f002:**
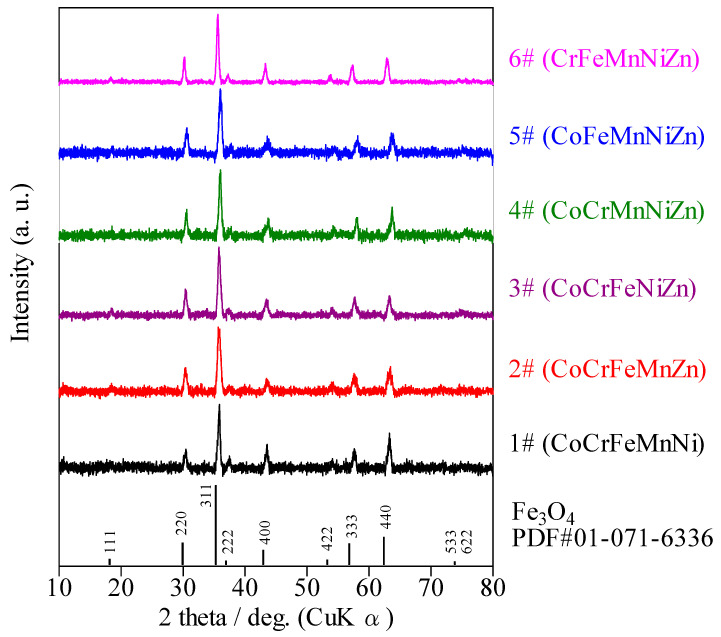
XRD patterns of the six HEOs and international center for diffraction data (ICDD) pattern for spinel Fe_3_O_4_.

**Figure 3 sensors-22-04233-f003:**
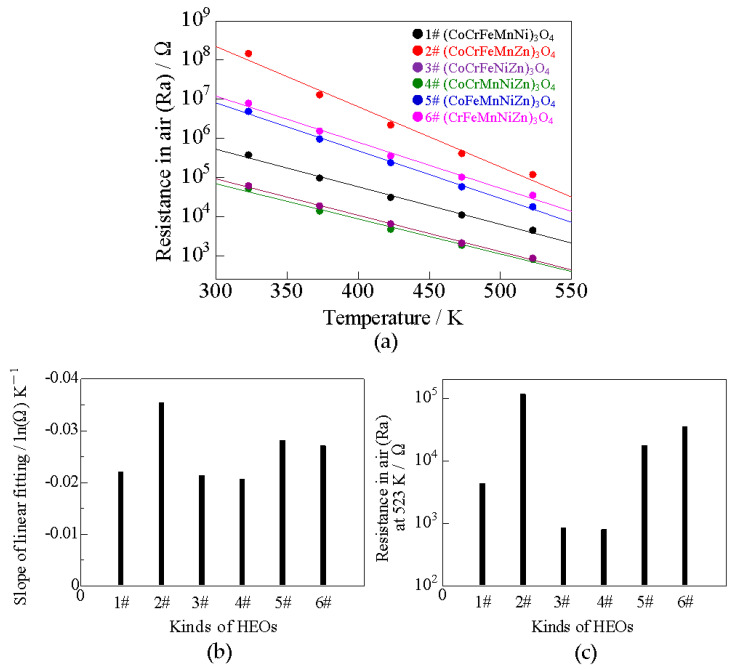
(**a**) Temperature dependence on Ra, (**b**) slope of linear fitting of (**a**), and (**c**) Ra at 523 K for the six HEOs.

**Figure 4 sensors-22-04233-f004:**
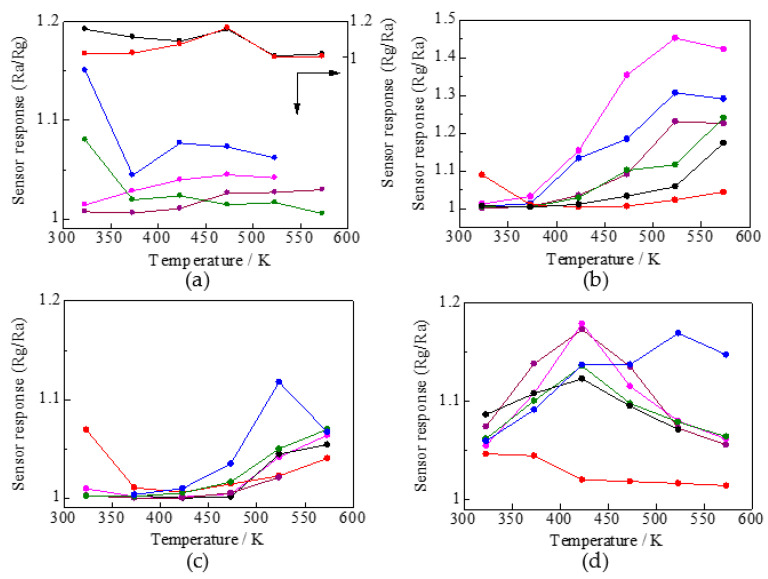
Temperature dependence of sensor response to (**a**) 7 ppm NO_2_, (**b**) 5000 ppm H_2_, (**c**) 3 ppm NH_3_, and (**d**) 3 ppm H_2_S. ● 1# (CoCrFeMnNi)_3_O_4_, ● 2# (CoCrFeMnZn)_3_O_4_, ● 3# (CoCrFeNiZn)_3_O_4_, ● 4# (CoCrMnNiZn)_3_O_4_, ● 5# (CoFeMnNiZn)_3_O_4_, ● 6# (CrFeMnNiZn)_3_O_4_.

**Figure 5 sensors-22-04233-f005:**
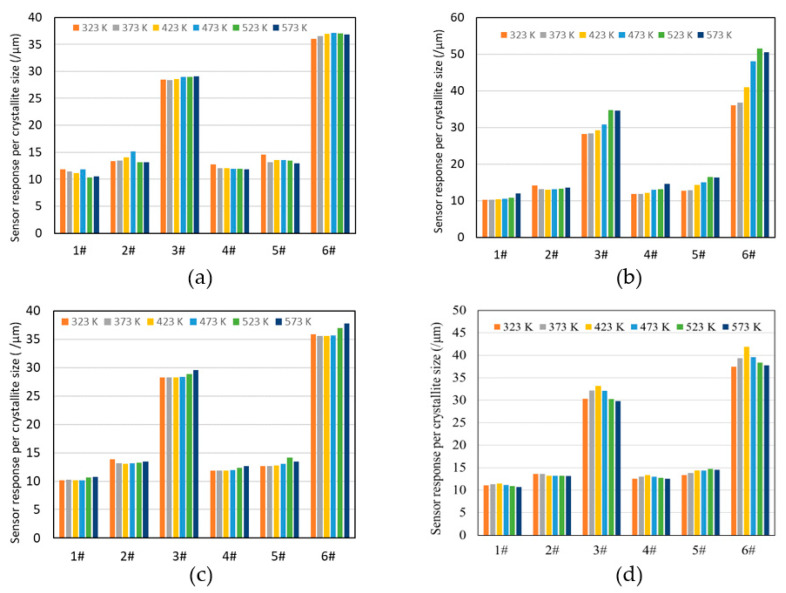
Sensor response per crystallite size for the six HEOs at each operating temperature to (**a**) 7 ppm NO_2_, (**b**) 5000 ppm H_2_, (**c**) 3 ppm NH_3_, and (**d**) 3 ppm H_2_S.

**Figure 6 sensors-22-04233-f006:**
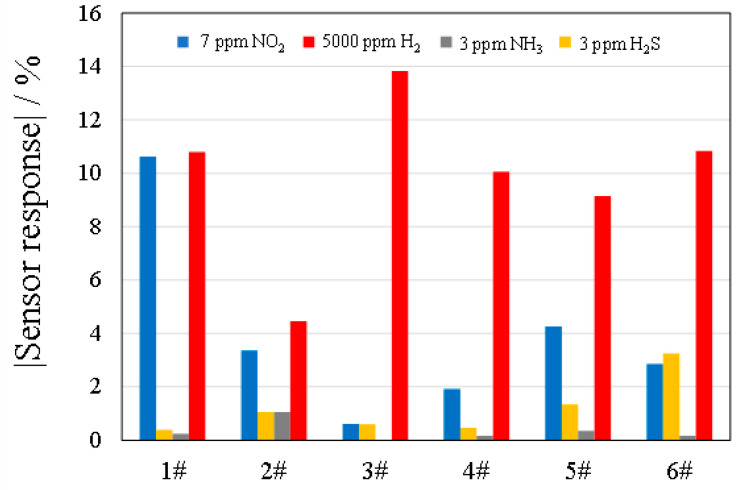
Sensor responses of the six HEOs at 373 K to 7 ppm NO_2_ (■), 5000 ppm H_2_ (■), 3 ppm NH_3_ (■), and 3 ppm H_2_S (■). Sensor response (%) is defined as |[Rg − Ra]/Ra × 100| for gas selectivity.

**Figure 7 sensors-22-04233-f007:**
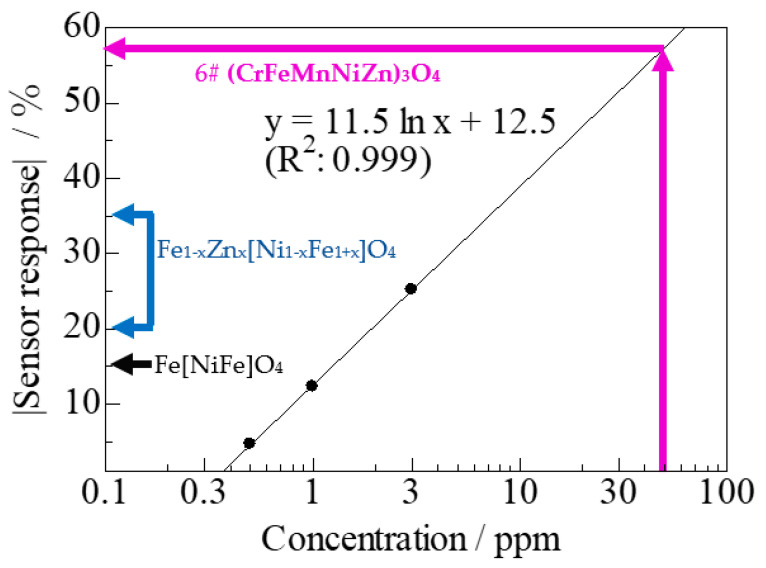
Sensor responses for spinel oxides at 423 K as a function of the H_2_S concentration. Pink arrow shows the sensor response for 6# HEO; the light blue arrow shows the detection range of the sensor response for Fe_1__−__x_Zn_x_[Ni_1__−__x_Fe_1__+__x_]O_4_, and the black arrow shows the sensor response for Fe[NiFe]O_4_ at 50 ppm H_2_S.

**Figure 8 sensors-22-04233-f008:**
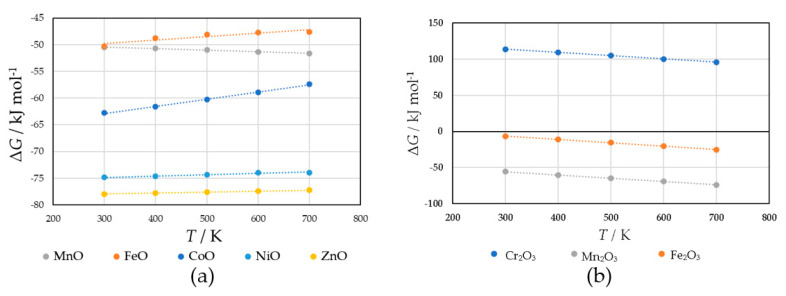
Ellingham diagram for (**a**) oxides with AO composition in the tetrahedral site and (**b**) oxides with B_2_O_3_ composition in the octahedral site.

**Table 1 sensors-22-04233-t001:** Crystallite size of each crystal plane for the six HEOs.

Crystal Plane	Crystallite Size/nm					
	1#	2#	3#	4#	5#	6#
111	175.8	155.8	28.8	227.4	140.3	48.6
220	20.5	18.2	29.5	27.2	17.3	25.3
311	22.8	15.1	20.2	20.7	16.0	23.1
222	250.6	153.8	81.2	148.8	274.3	20.8
400	31.5	66.2	15.1	29.9	48.6	23.0
422	207.2	135.2	66.6	163.7	69.2	46.6
511	49.4	44.1	23.0	26.3	31.5	21.5
440	27.8	27.8	18.9	34.6	36.2	16.2
Average	98.2	77.0	35.4	84.8	79.2	28.2

**Table 2 sensors-22-04233-t002:** Occupation factor, λ, in some spinels; λ = 0 denotes a normal spinel, and λ = 0.5 denotes an inverse spinel.

Spinel	A site	Mn^2+^	Fe^2+^	Co^2+^	Ni^2+^	Zn^2+^
B site	*d^m^*	*d^5^*	*d^6^*	*d^7^*	*d^8^*	*d^10^*
Cr^3+^	*d^3^*	0	0	0	0	0
Fe^3+^	*d^5^*	0.1	0.5	0.5	0.5	0
Mn^3+^	*d^4^*	N/A	N/A	N/A	N/A	0
Co^3+^	*d^6^*	N/A	N/A	N/A	0	0

**Table 3 sensors-22-04233-t003:** Cation migration tendency for the six HEOs. (**a**) Cation migration level for the six cations. (**b**) Cation migration level for the six HEOs.

(**a**)						
**Spinel**	**A Site**	**Mn^2+^**	**Fe^2+^**	**Co^2+^**	**Ni^2+^**	**Zn^2+^**
B site	*d^m^*	*d^5^*	*d^6^*	*d^7^*	*d^8^*	*d^10^*
Cr^3+^	*d^3^*	L	L	L	L	M
Fe^3+^	*d^5^*	L	L	L	L	H
Mn^3+^	*d^4^*	L	L	L	L	H
Co^3+^	*d^6^*	L	L	L	L	H
Ni^3+^	*d^7^*	L	L	L	L	M
(**b**)						
**Migration level**	**1#**	**2#**	**3#**	**4#**	**5#**	**6#**
	**without Zn**	**without Ni**	**without Mn**	**without Fe**	**without Cr**	**without Co**
H	0	3	2	2	3	2
M	0	1	2	2	1	2
L	20	12	12	12	16	12

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
