# Peer review of "Useful High-Entropy Source on Spinel Oxides for Gas Detection"

_sensors, 2022, doi:10.3390/s22114233_

Round 1

Reviewer 1 Report

The authors have prepared a series of high entropy metal oxides by solvothermal method and have evaluated their sensing properties. The paper main assumption is interesting, as concerns the varied effect of metal cations onto the sensing properties, but there are a number of issues to be examined by the authors before paper reconsideration. First of all, the structural analysis is only based onto the observation of the XRD patterns, without any determination of the crystal parameters. The latter seems particularly urgent, since the XRD patterns do not seem to show any relevant shift of the reflections depending on the composition, which is rather tricky. The concept of “cation migration level” is rather obscure and is anyway based onto a site occupancy which is not deeply explored in the paper, given the possibly of somewhat disordered nature of the prepared materials.  

The gas responses are rather modest and the paper does not report the dynamic response curves of the sensors.

Author Response

Dear Reviewer 1

Thank you for your valuable comments.

We responded to your valuable comments on the attached file.

Best regards,

T. Hashishin

Reviewer 2 Report

The paper is written well and can be accepted for publication  after adressing the following comments 

1. Authors has to mention the LOD (Limit of Detection) for their sample.

2. What is the basis of selecting element spinel oxides for gas sensing. 

Author Response

Dear Reviewer 2

Thank you for your valuable comments.

We responded to your valuable comments on the attached file.

Best regards,

T. Hashishin

Round 2

Reviewer 1 Report

The manuscript was properly revised and can be accepted in the present form.